# Applications of Advanced Ultrasound Technology in Obstetrics

**DOI:** 10.3390/diagnostics11071217

**Published:** 2021-07-06

**Authors:** Kwok-Yin Leung

**Affiliations:** Obstetrics and Gynaecology, Gleneagles Hong Kong, Hong Kong, China; ky@kyleung.org

**Keywords:** obstetrics, ultrasound, 3D, 4D, Doppler, artificial intelligence

## Abstract

Over the years, there have been several improvements in ultrasound technologies including high-resolution ultrasonography, linear transducer, radiant flow, three-/four-dimensional (3D/4D) ultrasound, speckle tracking of the fetal heart, and artificial intelligence. The aims of this review are to evaluate the use of these advanced technologies in obstetrics in the midst of new guidelines on and new techniques of obstetric ultrasonography. In particular, whether these technologies can improve the diagnostic capability, functional analysis, workflow, and ergonomics of obstetric ultrasound examinations will be discussed.

## 1. Introduction

Ultrasound is widely used in obstetric practice to detect fetal abnormalities with a view to provide prenatal opportunities for further investigations including genetic testing and discussion of management options. In 2010, International Societies of Ultrasound in Obstetrics and Gynecology (ISUOG) published the practice guidelines on the minimal and optional requirements for a routine mid-trimester ultrasound scan [1]. Recently, The American Institute of Ultrasound in Medicine (AIUM) suggests a detailed diagnostic second/third trimester scan for high-risk pregnancies [2], and fetal echocardiography for at-risk pregnancies [3]. ISUOG has published recent guidelines on indications and practice of targeted neurosonography [4,5]. Although the introduction of prenatal cell-free DNA-based screening for Down syndrome has changed the role of the first trimester scan, the latter should still be offered to women [6]. Around 50% of major structural abnormalities can be detected in the first trimester [7]. In addition, a recent study showed that a routine scan at around 36 weeks’ gestation can detect around 0.5% of previously undetected fetal abnormalities, as well as fetal growth restriction (FGR) [8].

The detection rate of fetal abnormalities varies, depending on anatomy survey protocol, ultrasound equipment and setting, among other factors [9]. A high-resolution ultrasound can facilitate a detailed diagnostic scan and a first-trimester scan and allow the detection of a small or subtle abnormality [10,11,12]. Although a detailed diagnostic scan is not required for all pregnant women, the indications include family history of congenital malformation, maternal age 35 or above, gestational diabetes mellitus, artificial reproduction technology, body mass index >= 30, teratogen, fetal nuchal translucency >= 3mm, and many other conditions [2]. In the midst of such increasing standards of obstetric ultrasound examination, there is a demand on improving the diagnostic capability, functional analysis, workflow, and ergonomics. Over the years, there have been several improvements in ultrasound technologies including high-resolution ultrasonography, linear transducer, radiant flow, three/four-dimensional (3D/4D) ultrasound, speckle tracking of the fetal heart, and artificial intelligence. The aim of this review is to evaluate the use of these advanced technologies in obstetrics.

## 2. High-Resolution Ultrasonography

High-resolution ultrasonography includes the use of a high-frequency transducer, and the means of enhancing image and signal processing including harmonic imaging (HI), spatial compound imaging (SCI), and speckle reduction imaging (SRI). Compared to a transducer with the low-frequency range (2 to 5 MHz), a transducer with the high-frequency range (5 to 9 MHz) can allow for improved resolution though with limited tissue penetration. HI, utilizing the physics of non-linear propagation of ultrasound through the body tissues, can produce high-resolution images with few artifacts. SCI, combining multiple lines of sight to form a single composite image at real-time frame rates, can reduce angle-dependent artifacts. The use of SRI can reduce speckles or disturbances that result from the echo, which is projected from an ultrasound transducer.

### 2.1. Fetal Echocardiography and Targeted Neurosonography

ISUOG recommends the use of the highest possible transducer frequency to perform fetal echocardiography with a view to improve the likelihood of detecting subtle heart defects, albeit at the expense of reduced acoustic penetration [10] (Figure 1a–d and Appendix A). The use of HI can improve the quality of ultrasound images, especially when the maternal abdominal wall is thick during the third trimester of pregnancy [11,13].

For a targeted neurosonographic examination, ISUOG recommends the use of high- resolution transvaginal transducers whenever possible [5]. An alternative is to use high-resolution transabdominal transducers with high frequency reaching 8–9 MHz [5]. The anatomy of the fetal brain is examined in details on a continuum of transverse, sagittal and coronal planes (Figure 2a–d, Appendix A).

### 2.2. Face and Neck

While the prenatal detection of cleft lip is high, the detection rate of subtle abnormalities of face such as low-set or posteriorly rotating ear, triangular face, down-slanting palpebral fissures, or a long and marked philtrum remains low [14,15]. These subtle abnormalities may be features of rare but severe genetic disorders such as 5p deletion syndrome or RASopathy, which require chromosome microarray analysis or targeted sequencing for RASopathy genes. As such, it is important to perform a detailed ultrasound scan to evaluate fetal face in fetuses especially if they have large NT, heart defects, or unusual findings [14,15]. High-resolution ultrasonography allows the clear visualization of facial profile, lens, nostrils, lips, maxilla, and ears (Figure 3a–d, Appendix A). Recently, a new sonographic sign, the ‘superimposed line’ sign, is suggested for evaluation of the secondary palate by assessment of the vomeromaxillary junction in the midsagittal view of the palate [16] (Figure 3a).

Larynx and its movement can be assessed by prenatal ultrasound (Figure 4 and Appendix A). In at-risk fetuses such as those with laryngeal atresia [17] and congenital diaphragmatic hernia, prenatal ultrasound allows systematic examination of the larynx, including vocal cords to detect laryngeal anomalies [17,18].

### 2.3. Early Pregnancy Scan

Transvaginal ultrasonography is essential in the assessment of pregnancy of unknown location, which can be due to early pregnancy, miscarriage, or ectopic pregnancy. It is important to avoid making a false-positive diagnosis of miscarriage by using transvaginal sonography, careful measurement of mean sac diameter and crown rump length, and using safe cut-off values of these measurements in defining miscarriage [19]. A recent study showed that amniotic sac sign (the presence of amniotic sac without a live embryo) is a reliable marker of miscarriage [20]. While the presence of an extrauterine gestational sac with yolk sac and/or embryo with or without cardiac activity is indicative of ectopic pregnancy, the presence of an inhomogeneous adnexal mass (‘blob’ sign) or extrauterine sac-like structure (‘bagel’ sign) is very suggestive of a tubal ectopic pregnancy [21]. In women with prior Caesarean section, ultrasound features of Caesarean scar pregnancy including low implantation of the gestational sac within or in close proximity to a Caesarean scar as well as classical signs of placenta accreta spectrum disorders should be looked out for [22,23].

### 2.4. First Trimester Scan

ISUOG and recently, AIUM published the practice guidelines on first-trimester fetal ultrasound scan [24,25]. High-resolution ultrasonography allows the early assessment of fetal anatomy [11] (Figure 5a–d, Appendix A) and fetal malformations [12]. Fetal heart can be examined in the late first trimester [26], particularly with the use of color Doppler (Appendix A). ISUOG recommends using high-frequency (6–12 MHz) transvaginal ultrasound to examine fetal brain, especially if the focus is in the posterior fossa and the maternal abdominal wall is thick [5].

### 2.5. Doppler Ultrasound

Doppler ultrasound is widely used in obstetrics. ISUOG has made recommendations on how to perform Doppler ultrasonography of the fetoplacental circulation [27]. It is a challenge to detect late-onset feral growth restriction (FGR). Although third-trimester–cerebroplacental ratio (CPR = middle cerebral artery pulsatile index/umbilical artery pulsatile index) is an independent predictor of stillbirth and perinatal mortality [28], CPR with or without adjustment for estimated fetal weight centile showed a low prediction rate for adverse perinatal outcome [29]. According to a meta-analysis, abnormal uterine artery (UtA) Doppler in the third trimester is useful in predicting perinatal death in suspected small-for-gestational age fetuses [30]. A recent prospective study suggested that cerebral–placental–uterine ratio (CPUR = CPR divided by mean UtA pulsatile index) detected FGR better than CPR or UtA Doppler alone [31].

### 2.6. Labour Ward Ultrasound

The use of intrapartum ultrasonography is increasing. It can be performed by using a portable machine equipped with a wide-sector and low-frequency (<4 MHz) transducer and batteries with a long life, and being quick to start up and recharge [32]. According to ISUOG practice guidelines [32], intrapartum ultrasound is indicated when there is slow progress or arrest of labor in the first or second stage. Recent studies showed that single ultrasound assessment of the fetal head station on admission in active phase or repeated measurements during active phase can predict the duration of labor and operative delivery in nulliparous women [33,34]. When the second stage of labor is prolonged, ultrasound can be used to assess fetal head position and station before considering or performing instrumental vaginal delivery [32]. Such assessment has a potential to predict mode of operative delivery and pregnancy outcomes [35]. Compared to clinical vaginal examinations, ultrasound assessment of the fetal head station and position is objective and reproducible [32,33,34], but assessment of cervical dilatation is limited when the dilatation is ≥ 4 cm and the membranes are ruptured [36].

Allowing detection of changes in tissue elasticity, elastography is a complementary technique to B-mode imaging, and it includes two methods, namely, shear-wave and strain elastography. A recent meta-analysis showed that the performance of cervical elastography was better than cervical length in the prediction of preterm delivery [37]. For the prediction of outcome of induction of labor, models based on inner cervical shear wave elastography and cervical length were more accurate than models based on the Bishop score [38].

## 3. Linear Transducer

With a high-frequency ultrasound, a linear transducer can produce high-resolution images of shallow structures and small parts. Unlike curved transducers, linear transducers produce a rectangular field of view with uniform beam density throughout all tissue levels and without divergence in deeper tissue. The use of a transabdominal linear transducer can enhance the examination of the spinal cord and conus medullaris in the midsagittal view of the spine [5]. Some abnormalities such as cataract [39] and laryngeal atresia [17] can be well demonstrated using a linear transducer.

A linear transducer can be used to examine fetal structures in the first trimester (Figure 6a–d). However, a linear transducer is not suitable for using if the structures of interest are deep or the maternal abdominal wall is thick. Although a linear transducer can allow the examination of the fetal cardiac anatomy at 11–13 weeks [40], it is the use of color flow mapping but not of a linear transducer that improves the examination [41].

## 4. Radiant Flow

Radiant flow shows the blood flow with a sense of depth by using a specific algorithm to convert the index of erythrocyte density in a certain area into a height index which is then superimposed on the initial coding of color, power Doppler, or high-definition flow [42]. Other advantages include reducing blood overflow and indicating the vessel with sharp edges. Special display produced by similar technologies include MicroFlow Imaging (Philips), MV-Flow, and LumiFlow (Samsung).

Radiant flow is used to show fast blood flow in the fetal heart and brachycephalic arteries [42] (Appendix A), as well as slow-blood flow in the neurovascular circulation [43] (Appendix A).

The fetal umbilical–portal venous system is complex. High-definition flow imaging (HDFI) has been used to assess the normal anatomy of this system or umbilical–portal–systemic venous shunts. Transverse and sagittal planes are used to examine the fetal umbilical–portal venous system (Appendix A). In a recent case report, the authors used HDFI and radiant flow imaging to clearly delineate the aberrant course of the ductus venosus returning to the coronary sinus [44].

## 5. 3D/4D Ultrasound

Over the years, new 2D modes (such as high-density power imaging), new 3D volume acquisition (such as Corpus callosum mode or matrix probe), and new analysis (such as semiautomated analysis) have been added in 3D/4D ultrasound examinations (Table 1). The use of 3D multiplanar/multislice analysis facilitates the assessment of normal and abnormal structures in standard planes. This can also facilitate the detection of subtle fetal defects [45]. The use of 3D rendered images can help counseling to the women when fetal malformations are found or reassure the at-risk women when normal fetal anatomy is found [45]. 3D/4D US is useful for the assessment of fetal brain, spine, face, heart, and other structures [45,46].

Examples in the assessment of fetal abnormalities
Cleft lip and palate: use gray-scale mode, after a 3D volume acquisition, perform multiplanar/multi-slice analysis and rendering techniques to assess the integrity of palate.Short-limbed and short-rib dysplasia: use gray-scale mode, after a 3D volume acquisition with skeletal mode, perform rendering techniques with skeletal mode to examine the long bones and ribs.Agenesis of ductus venosus: use high-density power imaging, after a 3D volume acquisition, perform multi-slice analysis to assess the precordial venous system.Cardiac outflow tract abnormalities: use color flow, after a STIC volume acquisition, perform multiplanar/multi-slice analysis in a cine-loop of cardiac cycle.Atrioventricular valve abnormalities: use matrix probe and gray-scale mode, real- time 4D cine-loop analysis to display the coronal view of atrioventricular valve.

### 5.1. 3D Neurosonography

In targeted neurosonography, a systematic assessment of the fetal brain is required. Although this assessment can be performed by a 2D ultrasound examination, a perfect midsagittal view may not be achieved at all times, thus affecting a proper assessment. ISUOG recommends using 3D ultrasound examination that can provide images of enhanced resolution by displaying thicker ‘slices’ of the brain and thus increasing the signal-to-background noise ratio on all three planes. In addition, multiplanar imaging correlation allows the display of perfectly aligned views on the three orthogonal planes [5]. To avoid shallowing by adjacent skull bones, it is important to acquire a 3D volume in a mid-sagittal plane through the sagittal suture. If the focus is on the anterior complex, the volume will be obtained from the anterior fontanelle or the anterior part of the sagittal suture [5] (Figure 7a). If the focus is on the posterior fossa and cerebellar vermis, the volume will be obtained from the posterior fontanelle or the posterior part of the sagittal suture with the ultrasound beam being almost perpendicular to the tentorium [47] (Figure 7b). A transvaginal and transabdominal approach is used when the fetal presentation is vertex and breech, respectively. Then, the midlines structures including corpus callosum, brain stem, and cerebellar vermis can be examined by multiplanar and multi-slice analysis [43,48,49]. An accurate measurement of corpus callosum and cerebellar vermis can be achieved.

After a 3D volume acquisition of the fetal spine at mid-sagittal plane, a rendered view of the fetal spine can be well displayed with various modes (Figure 8a,b). In addition, the coronal planes at the level of the vertebral bodies and/or posterior arches can be reconstructed on multiplanar analysis [5].

### 5.2. Spatiotemporal Image Correlation

Spatiotemporal Image Correlation (STIC) allows an automatic acquisition of a single 3D volume through slow sweep and subsequent analysis in a looped cine sequency of images in the multiplanar/multi-slice format and a rendered view. This can produce images in a standardized plane while minimizing the operation dependency of the ultrasound examination. The recent advances in gray scale and color Doppler post processing improves the display of ultrasound images. Using color Doppler with STIC in the glass-body mode can show normal and abnormal anatomy of the fetal heart and major vessels [46] (Figure 9, Appendix A). The matrix probe allows the rapid acquisition of an STIC volume, thus reducing the motion artifact and facilitating live 4D display [46]. In addition, the use of the matrix probe allows the simultaneous examination of two orthogonal planes of the fetal heart in the ‘biplane mode’. Additional use of image recognition software can help review cardiac structures in the standard planes [46]. The 3D rendered images are useful for counseling to parents. In addition, STIC volume can facilitate interdisciplinary consultation and teleconsultation [42].

### 5.3. 3D Ultrasound Examination of Face, Limbs, and Other Structures

While 2D ultrasound is a key tool for the detection of fetal anomalies, there are some anomalies such as facial clefts, micrognathia, and club foot in which 3D ultrasound may provide additional information or help counseling when such anomaly is suspected [9]. Compared to 2D ultrasound alone, combined approach of 2D and 3D ultrasound with multiplanar/multi-slice analysis can improve the detection or exclusion of cleft palate in fetuses with cleft lip [50] (Figure 10). Although 3D ultrasound is less sensitive for the detection of isolated cleft palate, a recent study showed that an accurate evaluation of palate requires 3D ultrasound examination with volume acquisition in a strictly axial transverse view of the palate [16]. The use of 3D ultrasound multiplanar analysis and 3D rendering view can facilitate the display of mid-sagittal plane of the fetal face and thus improve the accuracy of measurements of the mandible and the detection of micrognathia [51].

Three-dimensional (3D) rendering technology with skeletal mode can display skull, vertebrae, ribs, long bones and fingers [52] (Figure 11a,b and Appendix A). Prenatal assessment of the ribs and vertebral pattern can be performed by 3D ultrasound with skeletal mode (Figure 8a,b), albeit it is not a routine practice. A review of 39 studies including 75,018 healthy subjects and 6130 subjects with structural or chromosomal anomalies or adverse outcome showed an association between cervical ribs and other structural anomalies including esophageal atresia and anorectal malformation [53]. Abnormalities such as craniosynostosis [26,54], and extra ribs can be shown.

It is difficult to visualize esophagus on 2D ultrasound examination. The use of 3D ultrasound with multiplanar analysis and Crystal Vue rendering may make the visualization possible [55]. Three-dimensional (3D) volumes are acquired from a midsagittal section of the thorax and upper abdomen with the fetus lying in supine position.

While 2D ultrasound examination with gray scale and color flow is the standard for the antenatal diagnosis of placenta accreta spectrum disorders [22], 3D ultrasound with power Doppler and multiplanar analysis permits an accurate assessment of the placenta-bladder interface, and the degree of bladder invasion by the placenta [56]. Three-dimensional (3D) rendered images can be used for patient counseling [56].

### 5.4. 3D Printing

With advances in 3D ultrasound, the derived ultrasound data can be used for 3D printing of physical models of whole fetuses [57] and the fetal face [58]. A recent trial showed that the use of 3D-printed fetal facial models resulted in greater increases in maternal–fetal attachment in the third trimester than the use of ultrasonography only [58]. Whether this can be translated into better pregnancy outcomes needs further studies. In addition, a 3D-printed spina bifida model can be beneficial for surgical rehearsal prior to a fetoscopic repair [59].

With advances in STIC, the derived data can be used for 3D printing of the fetal heart, which is a fast-moving structure [60]. In a recent case report, the authors found that the 3D model was useful in showing the complex anatomy of fetal transposition of great arteries and in providing prenatal parental counseling [61].

Previously, after acquisition of a 3D/STIC volume dataset, a number of post-processing steps are required to convert it from Cartesian.vol file through segmentation, refinement, and optimization to a STL (Standard Triangle Language) file, the industry standard file type for 3D Printing [60]. These steps take a long time, and whether the final produced STL file is good enough for 3D printing is not certain before processing. With recent advances in ultrasound technology, a 3D/STIC volume dataset can be directly exported from the ultrasound machine as an STL file that is ready for viewing on a personal computer using common software as well as for 3D printing (Figure 12).

## 6. FetalHQ

FetalHQ, a novel heart and vascular analysis software, can allow assessment of the fetal heart shape, size, and contractibility by using speckle tracking to analyze the motion of multiple points of the fetal heart [62] (Figure 13). The global sphericity index (SI) is a simple measurement of cardiac contractility, and it is equal to (end-diastolic mid-basal–apical length)/transverse length [63]. For 24-segment sphericity index, SI is computed for each of the 24 end-diastolic transverse segments, which are distributed from the base to the apex of each ventricle, as well as the end-diastolic mid-basal–apical length [62].

This 24-segment sphericity index is a comprehensive method to assess the shape of ventricular chambers [62]. The SI for each segment was independent of gestational age and fetal biometry. The SI of the right ventricle was lower than that of the left ventricle for segments 1–18. This index can be used when discordance between the size of the atrial and/or ventricular cardiac chambers is found. Abnormal SI values are found in the fetuses with cardiac abnormalities such as coarctation of aorta, pulmonary stenosis, and fetal growth restriction [62]. Abnormal SI values are associated with increased risk of perinatal complications and childhood and/or adult cardiovascular disease [64].

While the initial results are promising, a recent review of 23 studies showed conflicting results concerning the development of strain and strain rate during gestation [65]. Large longitudinal cohort studies with a standard protocol are needed to obtain reference values for fetal cardiac deformation in uncomplicated pregnancies [65]. A recent systematic review also showed heterogeneous results concerning gestational age and Doppler profiles. Large prospective longitudinal cohort studies are required to assess the clinical significance of deformation measurements of the fetal heart in growth restricted fetuses and normal fetuses [66].

## 7. Artificial Intelligence

Machine learning, in particular deep learning, allows ultrasound image recognition and thus facilitates the automatic identification and measurement of fetal biometry [67]. It is a branch of artificial intelligence (AI). In obstetric ultrasonography, the automation of measurements of fetal biometry is a potentially useful tool to increase the reliability and reproducibility of measurements as compared to manual measurements [68]. In addition, it can reduce scanning time [68] and work-related fatigue and musculoskeletal disorders [69].

With automatic image recognition technology applied on a frozen 2D ultrasound image, auto measurement of fetal biometry including biparietal diameter (BPD), head circumference (HC), abdominal circumference (AC), and femur length (FL) becomes feasible. A study showed a success rate of 91.43% and 100% for auto measurement of HC and BPD, respectively [67]. Although the inaccuracy for the plane acceptance check for head parameters was 12.9% [67], such inaccuracy can be corrected by fine-tuning of the caliper placement manually. In another study, manual adjustment of caliper position was not required in about two-thirds of cases for HC and FL measurements, but it was required in more than 80% for the measurement of AC [68]. Auto measuring AC is more difficult than measuring HC because of the low contrast between the abdomen and surrounding tissues and the large variability in abdominal shape and appearance [68]. The accuracy of the auto measurement of HC, AC, and FL was high, and it compared well with previously published manual-to-manual agreement, but the auto measurements had a tendency to underestimate biometry, which requires further improvements in the algorithm [68].

With automatic image recognition technology applied on an acquired 3D ultrasound volume of the fetal head from the BPD plane, SonoCNS allows auto measurement of fetal biometry including BPD, HC, atrium of the posterior horn of the lateral ventricle (Vp), transcerebellar diameter (TCD), and cisterna magnum (CM) [70] (Figure 14). A recent study showed that this 3D automated technology reliably identified and measured BPD and HC but was less so for TCD, CM, and Vp [70]. Further optimization of this automated technology is required.

Fetal Intelligent Navigation Echocardiography (FINE) applied on a STIC volume using “intelligent navigation” technology allows the automatic display of nine standard fetal echocardiography views [71]. This can simplify fetal cardiac examinations, reduce operator dependency, and help detect congenital heart defects [71].

During a real-time 2D morphology scan, identifying and interpreting fetal standard scan planes are highly complex tasks. With automatic image processing technology [46,70], these tasks can be assisted by providing feedback or guidance to an ultrasound operator on whether a correct standard scan plane of fetal anatomy is obtained, whether all parts of anatomy are checked, and whether unusual findings on a standard plane are identified [72]. The operator can use this technology as a second pass or confirmation to improve diagnostic accuracy [70]. This can also allow audit and quality improvement [73].

Based on deep learning, image segmentation is an image processing method that can automatically recognize the location and size of an object in pixels. However, accurate segmentation of most anatomical structures in medical ultrasound is limited by the low contrast between the target and background of the images [74]. To improve the segmentation performance of the thoracic wall in fetal ultrasound videos, a novel model-agnostic method using deep learning techniques in processing time-series information of ultrasound videos and the shape of the thoracic wall was proposed [75]. Accurate segmentation can assist ultrasonographers with identifying the thoracic area and its orientation, and it has the potential to build AI-based diagnostic support models to assess four-chamber view [75].

There are emerging studies on the application of artificial intelligence in obstetric scan. It is feasible to use 3D ultrasound to automatic measure thymic volume [76]. Two-dimensional (2D) placental sonographic images can be screened for lacunae, which is a feature of PAS [77]. Preliminary results are encouraging. Further improvement of algorithm and technology are required prior to using AI applications in clinical practice.

## 8. Conclusions

The use of high-resolution ultrasonography can facilitate detailed diagnostic ultrasonography, in particular, fetal echocardiography and targeted neurosonography, in at-risk pregnancies, as suggested by the recent guidelines. The use of radiant flow can improve the display, especially in complex cardiac or vascular structures. The use of 3D/4D ultrasound may help in the prenatal diagnosis and counseling of some fetal abnormalities. Select use of linear transducer may enhance the diagnostic capabilities of some superficial anomalies. Speckle tracking of the fetal heart can allow assessment of fetal heart shape, size and contractibility, and further studies are required to assess its clinical effectiveness. At present, automated tools for simple task such as measurement of fetal growth biometry are a good assistant to routine ultrasonography. Further refinement of automated algorithm is required, especially for complex tasks, to improve the workflow.

## Figures and Tables

**Figure 1 diagnostics-11-01217-f001:**
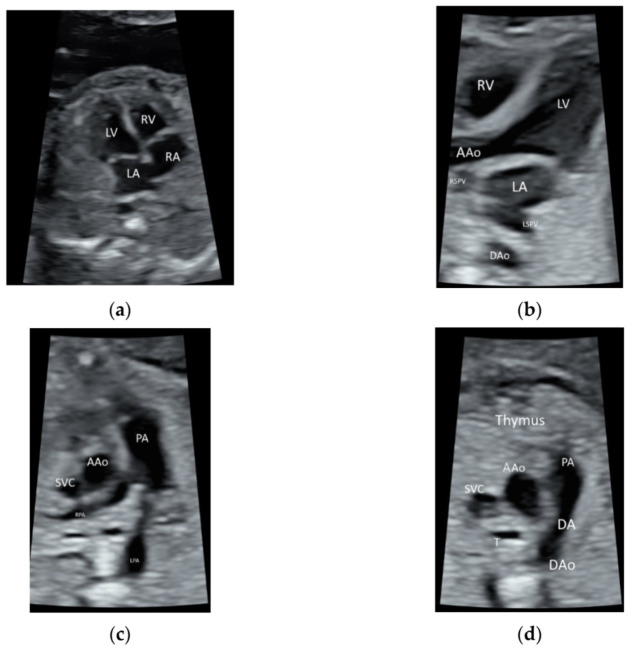
High-resolution ultrasonography of the fetal heart at 20 weeks’ gestation showing (**a**) a four-chamber view showing right atrium (RA), left atrium (LA), right ventricle (RV), and left ventricle (LV), (**b**) five-chamber view showing ascending aorta (AAo) arising from the left ventricle, the right and left superior pulmonary veins (RSPV, LSPV) enter the left atrium (LA), and descending aorta (DAo) behind the LA (**c**) Three-vessel view showing the PA dividing into the left (LPA) and right (RPA) PA, AAo, and the superior vena cava (SVC), (**d**) three-vessel and trachea view showing PA with the ductal branch (DA) joining the DAo, AAo, SVC, and trachea (T); Thymus is anterior to the three vessels.

**Figure 2 diagnostics-11-01217-f002:**
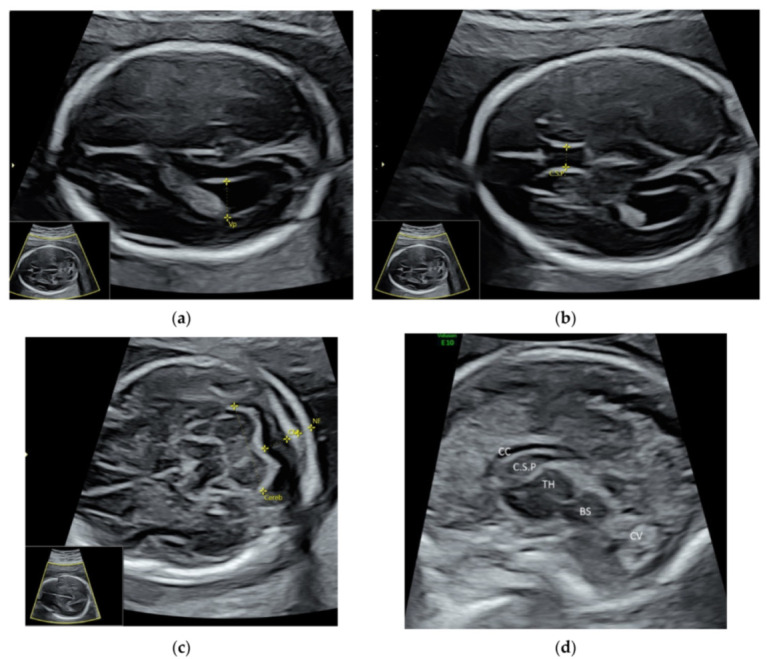
High-resolution ultrasonography of the fetal brain at 20 weeks’ gestation: transverse views showing (**a**) posterior horn of the lateral ventricle (Vp), (**b**) cavum septi pellucidi (C.S.P.), (**c**) cerebellum (Cereb), Cisterna magna (CM), nuchal fold (NF), and sagittal view showing (**d**) corpus callosum (CC), thalamus (TH), brain stem (BS), and cerebellar vermis (CV).

**Figure 3 diagnostics-11-01217-f003:**
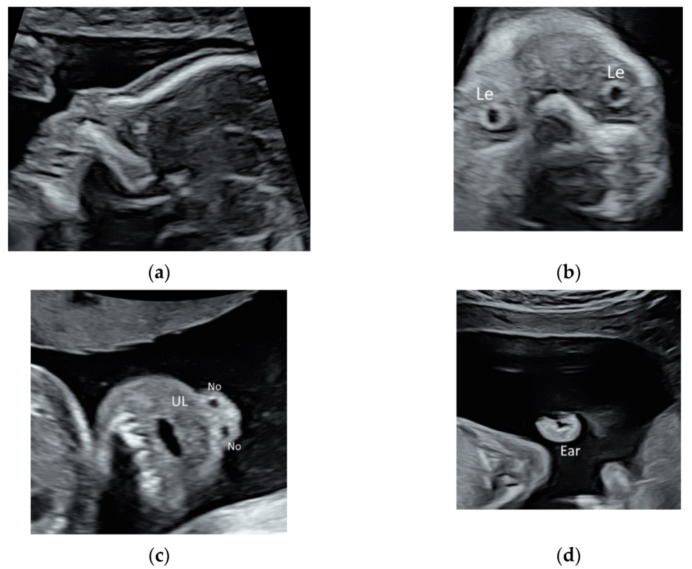
High-resolution ultrasonography of the fetal face at 20 weeks’ gestation: (**a**) mid-sagittal view showing facial profile, (**b**) coronal view showing both lens (Le), (**c**) coronal view showing the upper lip (UL) and two nostrils (No), and (**d**) sagittal view showing the ear.

**Figure 4 diagnostics-11-01217-f004:**
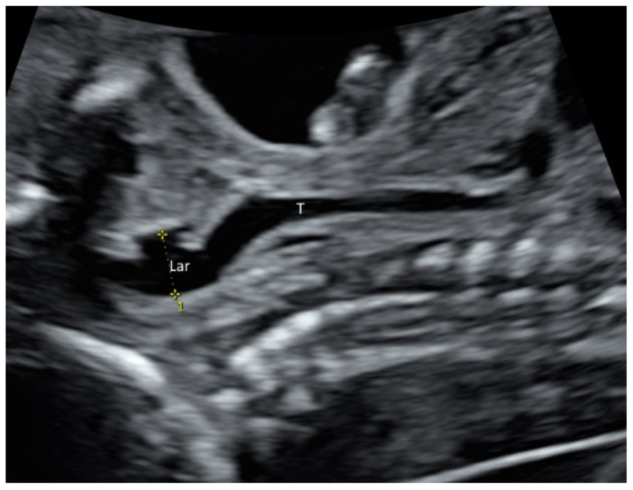
High-resolution ultrasonography of the fetal neck at 21 weeks’ gestation: sagittal view showing larynx (Lar) and trachea (T).

**Figure 5 diagnostics-11-01217-f005:**
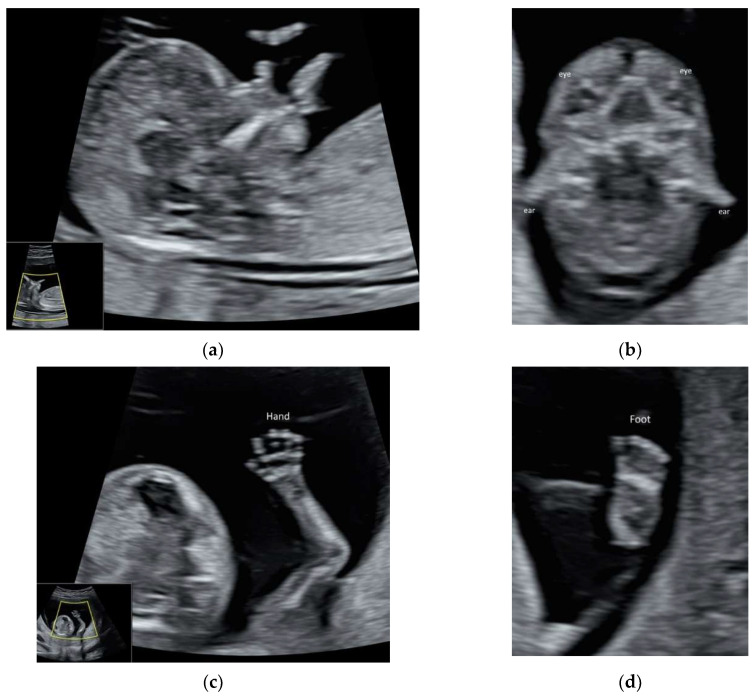
High-resolution ultrasonography of the fetus at 13 weeks’ gestation: (**a**) mid-sagittal view showing head, neck, and facial profile, (**b**) coronal view showing both eyes and ears, (**c**) the hand with five fingers, and (**d**) foot.

**Figure 6 diagnostics-11-01217-f006:**
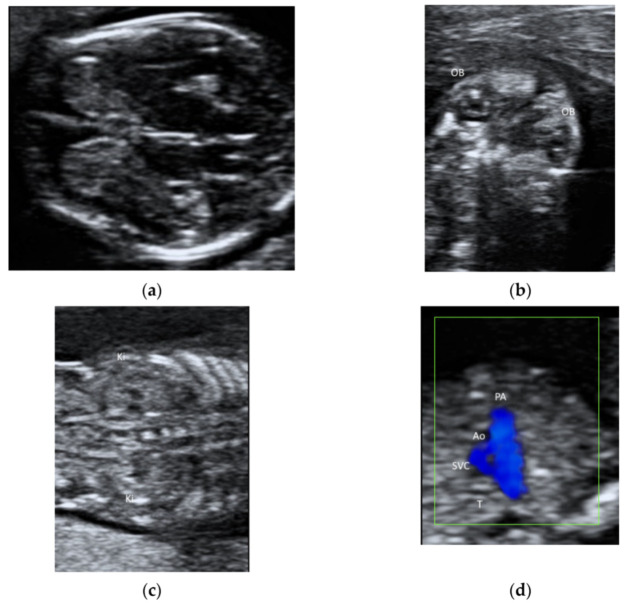
Ultrasonography of a fetus at 13 weeks’ gestation by a transabdominal high-frequency linear transducer: (**a**) transverse view of fetal brain, (**b**) coronal view of face showing both orbits (OB), (**c**) coronal view of abdomen showing both kidneys (Ki) on either side of the spine, and (**d**) the three-vessel trachea transverse view with color Doppler showing pulmonary artery (PA), aorta (Ao), superior vena cava (SVC), and trachea (T).

**Figure 7 diagnostics-11-01217-f007:**
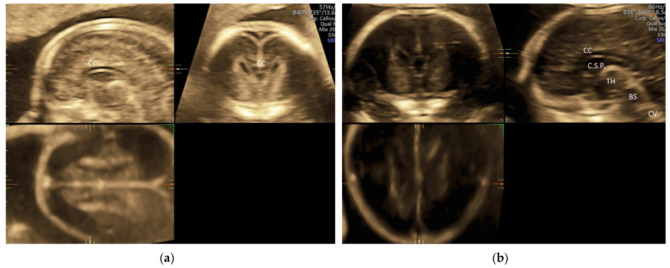
Three-dimensional ultrasonography of fetal brain at 20 weeks’ gestation: (**a**) multiplanar analysis after a volume acquisition with corpus callosum mode through the anterior part of the sagittal suture showing corpus callosum (CC), and (**b**) multiplanar analysis after a volume acquisition through the posterior fontanelle showing corpus callosum (CC), cavum septi pellucidi (C.S.P.), thalamus (TH), brainstem (BS), and cerebellar vermis (CV).

**Figure 8 diagnostics-11-01217-f008:**
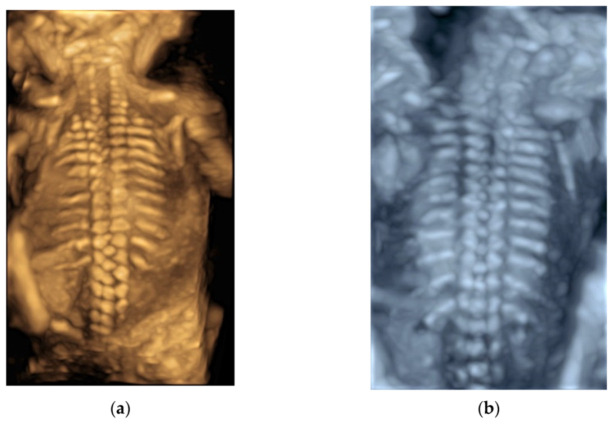
Three-dimensional rendered views of fetal spine at 20 weeks’ gestation after a volume acquisition with skeletal mode: (**a**) usual mode, and (**b**) X-ray mode.

**Figure 9 diagnostics-11-01217-f009:**
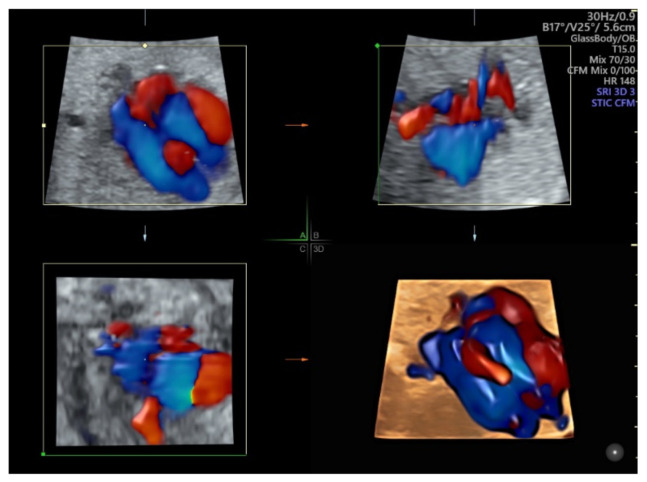
Color Doppler with spatiotemporal image correlation in the glass-body mode showing multiplanar view and a rendered image of a normal fetal heart at 20 weeks’ gestation.

**Figure 10 diagnostics-11-01217-f010:**
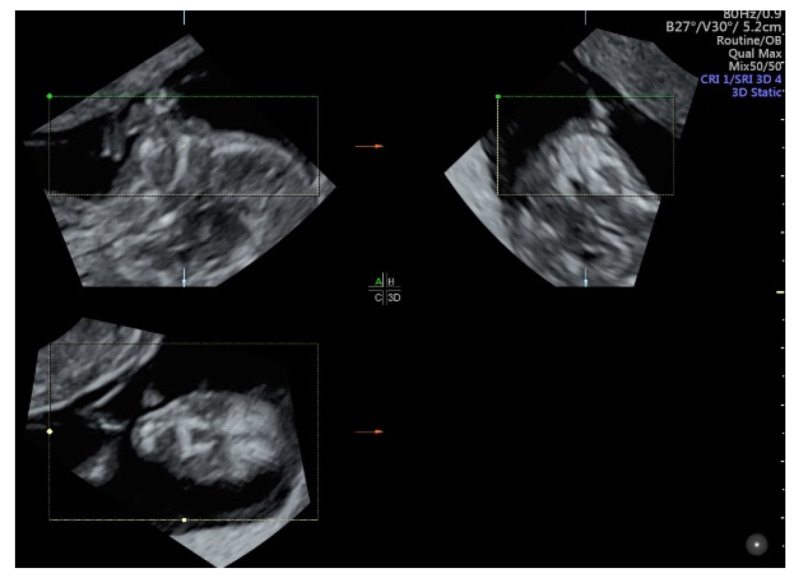
Three-dimensional ultrasound assessment of the fetal face at 13 weeks’ gestation showing multiplanar views of lip and palate (reference dot).

**Figure 11 diagnostics-11-01217-f011:**
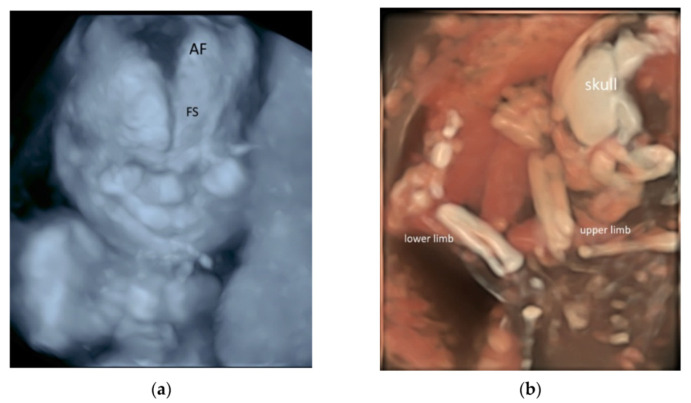
Three-dimensional rendered images of the fetal skeleton at 20–22 weeks’ gestation showing: (**a**) X-ray mode of the skull bone with frontal suture (SS) and anterior fontanelle (AF), and (**b**) HD skeletal mode of the skull, bones of the upper and lower limbs.

**Figure 12 diagnostics-11-01217-f012:**
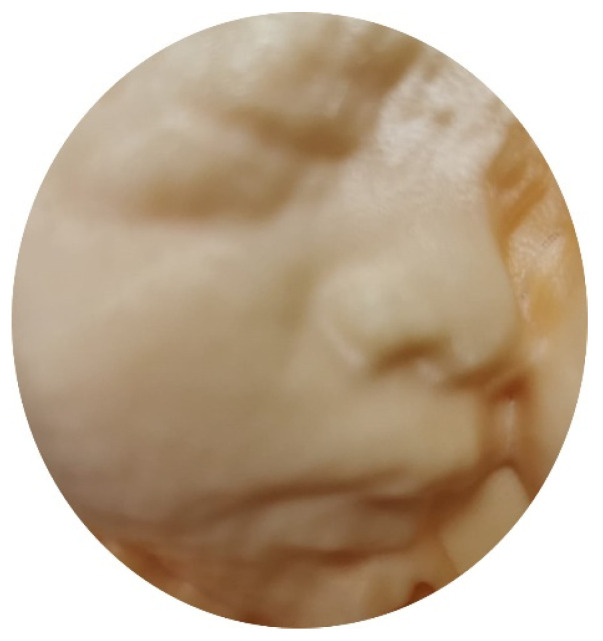
A physical model of three-dimensional printing of the fetal face.

**Figure 13 diagnostics-11-01217-f013:**
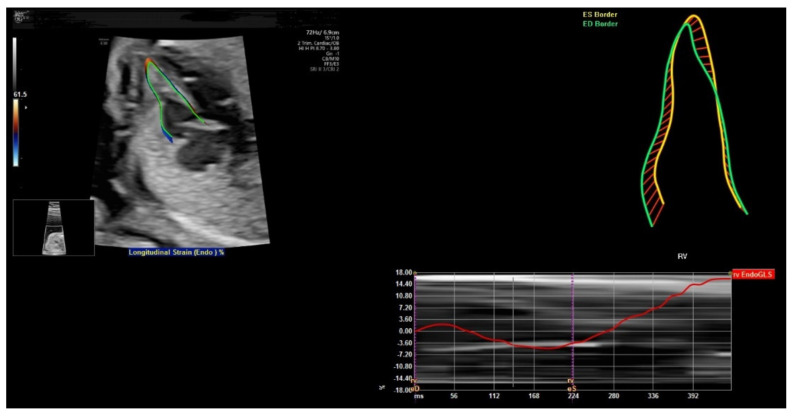
FetalHQ assessment of the fetal heart shape, size, and contractibility by using speckle tracking to analyze the motion of multiple points of the fetal heart at 20 weeks’ gestation.

**Figure 14 diagnostics-11-01217-f014:**
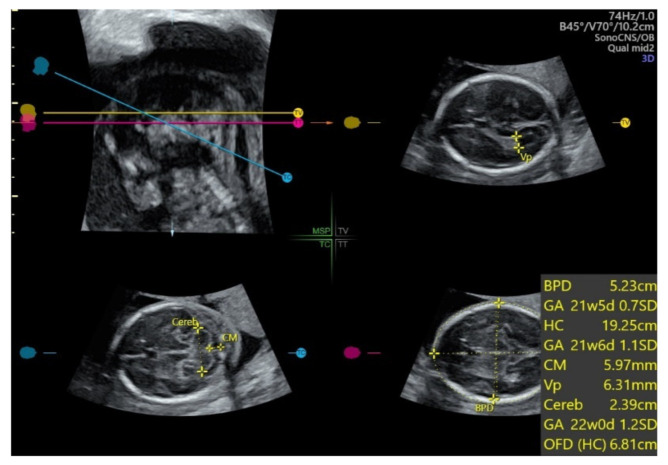
SonoCNS, after volume acquisition of the fetal brain at biparietal diameter plane at 21 weeks gestation, showing auto measurement of biparietal diameter (BPD), head circumference (HC), atrium of the posterior horn of the lateral ventricle (Vp), transcerebellar diameter (TCD), and cisterna magnum (CM).

**Table 1 diagnostics-11-01217-t001:** Commonly used scanning mode, volume acquisition, and analysis for three-/four-dimensional (3D/4D) ultrasound examinations.

Mode	Volume Acquisition	3D/4D Analysis
Gray scale	3D: different modes	Multiplanar
Color flow	4D	Multislice
Power doppler	STIC ^1^	Rendered view: different modes
High-density power imaging	Matrix probe	Cine loop
B-flow		Semi-automatic analysis
		Volume measurement
		Power Doppler measurements

^1^ Spatiotemporal image correlation.

## Data Availability

Data is contained within the article or Appendix A.

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
