# Peer review of "Applications of Advanced Ultrasound Technology in Obstetrics"

_diagnostics, 2021, doi:10.3390/diagnostics11071217_

Round 1

Reviewer 1 Report

The aim of the manuscript was to evaluate the value and diagnostic efficacy of different ultrasound modes in various foetal abnormalities. This review is potentially very interesting to readers. 
Several educational ultrasonography figures and videsos increase the value of the paper.
I have only minor remarks:
1. I am not able to find “Table 4” (line 116)
2. Table 1 should include application examples in foetal abnormalities
3. Is it possible to include the model-agnostic method for thoracic wall segmentation in fetal ultrasound?

Author Response

Thanks a lot for reviewing my manuscript and all the useful comments. My response is: 

1. I have deleted the phase 'Table 4', and revised the legend as: '

'Figure 6. Ultrasonography of a fetus at 13 weeks’ gestation by a transabdominal high-frequency linear transducer: (a) transverse view of fetal brain, (b) coronal view of face showing both orbits (OB), (c) coronal view of abdomen showing both kidneys (Ki) on either side of the spine, and (d) the three-vessel-trachea transverse view with color Doppler showing pulmonary artery (PA), aorta (Ao), superior vena cava (SVC) and trachea (T).'

2. In Table 1, I have added examples as follows:

'Examples in the assessment of fetal abnormalities

a. Cleft lip and palate: use gray scale mode, after a 3D volume acquisition, perform multiplanar/ multi-slice analysis and rendering techniques to assess the integrity of palate.

b. Short-limbed and short-rib dysplasia: use gray scale mode, after a 3D volume acquisition with skeletal mode, perform rendering techniques with skeletal mode to examine the long bones and ribs.

c. Agenesis of ductus venosus: use high-density power imaging, after a 3D volume acquisition, perform multi-slice analysis to assess the precordial venous system

d. Cardiac outflow tract abnormalities: use color flow, after a STIC volume acquisition, perform multiplanar/ multi-slice analysis in a cine-loop of cardiac cycle.

e. Atrioventricular valve abnormalities: use matrix probe and gray scale mode, real- time 4D cine-loop analysis to display the coronal view of atrioventricular valve'

3. In session 7 on Artificial Intelligence, I have added a paragraph: 

'Based on deep learning, image segmentation is an image processing method that can automatically recognize the location and size of an object in pixels. However, accurate segmentation of most anatomical structures in medical ultrasound is limited by the low contrast between the target and background of the images [77]. To improve the segmentation performance of the thoracic wall in fetal ultrasound videos, a novel model-agnostic method using deep learning techniques in processing time-series information of ultrasound videos, and the shape of the thoracic wall was proposed [78]. Accurate segmentation can assist ultrasonographers identify the thoracic area and its orientation, and has the potential to build AI-based diagnostic support models to assess four-chamber view [78].'

Reviewer 2 Report

This is a well designed and presented study. it has a well-structured formation. and the new advances in obstetric ultrasound are presented in a clear and precise way. However I believe that some more information regarding the early pregnancy scan , the doppler scan and the labour ward scan should be added. 

Therefore a paragraph regarding the scan prior to nuchal translucency, a paragraph with regard to doppler scan and a paragraph with regard to labour ward scan prior to delivery for the assessment of labour should be added. Otherwise it is a very nice and informative study, 

Author Response

Thanks a lot for reviewing my manuscript and all the useful comments. My response is: 

In Section 2 on high resolution ultrasonography, I have added paragraphs on early pregnancy scan, Doppler scan  and labour ward scan.

'2.3 Early pregnancy scan

Transvaginal ultrasonography is essential in the assessment of pregnancy of unknown location which can be due to early pregnancy, miscarriage, or ectopic pregnancy. It is important to avoid making a false-positive diagnosis of miscarriage by using transvaginal sonography, careful measurement of mean sac diameter and crown rump length, and using safe cut-off values of these measurements in defining miscarriage [19]. A recent study showed that amniotic sac sign (the presence of amniotic sac without a live embryo) is a reliable marker of miscarriage [20]. While the presence of an extrauterine gestational sac with yolk sac and/or embryo, with or without cardiac activity is indicative of ectopic pregnancy, the presence of an inhomogeneous adnexal mass ('blob' sign) or extrauterine sac-like structure ('bagel' sign) is very suggestive of a tubal ectopic pregnancy [21]. In women with prior Caesarean section, ultrasound features of Caesarean scar pregnancy including low implantation of the gestational sac within or in close proximity to a Caesarean scar as well as classical signs of placenta accreta spectrum disorders should be looked out for [22,23].'

'2.5 Doppler ultrasound

Doppler ultrasound is widely used in obstetrics. ISUOG has made recommendations on how to perform Doppler ultrasonography of the fetoplacental circulation [27]. It is a challenge to detect late-onset feral growth restriction (FGR). Although third-trimester -cerebroplacental ratio (CPR= middle cerebral artery pulsatile index/ umbilical artery pulsatile index) is an independent predictor of stillbirth and perinatal mortality [28], CPR with or without adjustment for estimated fetal weight centile showed a low prediction rate for adverse perinatal outcome [29]. According to a meta-analysis, abnormal uterine artery (UtA) Doppler in the third trimester is useful in predicting perinatal death in suspected small-for-gestational age fetuses [30]. A recent prospective study suggested that cerebral-placental-uterine ratio (CPUR = CPR divided by mean UtA pulsatile index) detected FGR better than did CPR or UtA Doppler alone [31].'

 '2.6 Labour ward ultrasound

The use of intrapartum ultrasonography is increasing. It can be performed by using a portable machine equipped with a wide-sector and low-frequency (<4MHz) transducer and batteries with a long life, and being quick to start up and recharge [32]. According to ISUOG practice guidelines [32], intrapartum ultrasound is indicated when there is slow progress or arrest of labor in the first or second stage. Recent studies showed single ultrasound assessment of the fetal head station on admission in active phase or repeated measurements during active phase can predict duration of labor and operative delivery in nulliparous women [33,34]. When second stage of labour is prolonged, ultrasound can be used to assess fetal head position and station before considering or performing instrumental vaginal delivery [32]. Such assessment has a potential to predict mode of operative delivery and pregnancy outcomes [35]. Compared to clinical vaginal examinations, ultrasound assessment of the fetal head station and position is objective and reproducible [32-34], but assessment of cervical dilatation is limited when the dilatation is  ≥ 4 cm and the membranes are ruptured [36]. 

Allowing detection of changes in tissue elasticity, elastography is a complementary technique to B-mode imaging, and include two methods, namely, shear-wave and strain elastography. A recent meta-analysis showed that the performance of cervical elastography was better than cervical length in the prediction of preterm delivery [37]. For the prediction of outcome of induction of labour, models based on inner cervical shear wave elastography and cervical length was more accurate than models based on the Bishop score [38].'